# Selecting predictive biomarkers from genomic data

**Florian Frommlet** [1]*, **Piotr Szulc**[2], **Franz König**[1], **Malgorzata Bogdan**[2,3]

**1** Department of Medical Statistics, CEMSIIS, Medical University of Vienna, Vienna, Austria, **2** Institute of Mathematics, University of Wroclaw, Wroclaw, Poland, **3** Department of Statistics, Lund University, Lund, Sweden

* Florian.Frommlet@meduniwien.ac.at

## Abstract

Recently there have been tremendous efforts to develop statistical procedures which allow to determine subgroups of patients for which certain treatments are effective. This article focuses on the selection of prognostic and predictive genetic biomarkers based on a relatively large number of candidate Single Nucleotide Polymorphisms (SNPs). We consider models which include prognostic markers as main effects and predictive markers as interaction effects with treatment. We compare different high-dimensional selection approaches including adaptive lasso, a Bayesian adaptive version of the Sorted L-One Penalized Estimator (SLOBE) and a modified version of the Bayesian Information Criterion (mBIC2). These are compared with classical multiple testing procedures for individual markers. Having identified predictive markers we consider several different approaches how to specify subgroups susceptible to treatment. Our main conclusion is that selection based on mBIC2 and SLOBE has similar predictive performance as the adaptive lasso while including substantially fewer biomarkers.

**Data Availability Statement:** The manuscript only makes use of simulated data. If required we will make the R Code used for simulations available.

**Funding:** PS and FK were supported by the European Union's 7th Framework Programme for research, technological development and

## Introduction

In the development of personalized medicine one important task is to determine subgroups of patients with a certain disease which might differ in their benefit from a specific treatment. Biomarkers which allow to predict the outcome of a treatment in comparison to a control group are usually called predictive biomarkers. In contrast biomarkers which predict the outcome irrespective of treatment are called prognostic. There exists a large body of literature concerning issues of study design and statistical analysis involved in determining predictive biomarkers; several recent books and reviews are available [1–5].

Subgroup analyses incorporating biomarkers are a routine part when conducting clinical trials to evaluate whether treatment effects are homogeneous across study population. To develop targeted therapies, patient subgroups are typically defined by genetic or proteomic biomarkers. The importance of identifying subgroups with a better benefit/risk profile has been acknowledged by regulatory agencies, for example both the U.S. Food and Drug Administration (FDA) and European Medicines Agency (EMA) have published guidelines on the

demonstration under Grant Agreement no 602552, https://ec.europa.eu/growth/sectors/space/research/fp7_en PS and FK were co-financed by the Polish Ministry of Science and Higher Education under Grant Agreement 2932/7.PR/2013/2. https://www.gov.pl/web/science MB gratefully gratefully acknowledges the support by the grant Nr 2016/23/B/ST1/00454 of the Polish National Center of Science. https://ncn.gov.pl/?language=en The funders had no role in study design, data collection and analysis, decision to publish, or preparation of the manuscript.

**Competing interests:** The authors have declared that no competing interests exist.

investigation of subgroups in clinical trials [6, 7]. A broad range of clinical trial designs using both prognostic and predictive enrichment strategies have been proposed [2, 3, 8–12] including adaptive enrichment designs [13, 14]. For an overview on exploratory and confirmatory methods we refer to the review of Ondra et al. [2].

Predictive biomarkers can be identified by modelling interactions between treatment effect and biomarkers. To this end one might distinguish between machine learning approaches (including support vector machines and regression tree based approaches) and more classical statistical approaches based on parametric regression models suited to the type of outcome, like linear regression for quantitative measurements, logistic regression in case of binary outcomes and Cox regression for survival (see [3] for a brilliant and comprehensive overview of existing methods). In this article we will focus on parametric regression where prognostic effects of biomarkers correspond to main effects in the regression model, whereas predictive effects are modelled by interaction terms between treatment and biomarkers. Based on the final model for each patient a predictive index can be calculated evaluating whether the patient may benefit from a new treatment or not. This index is also referred to as predicted individual treatment effect (PITE) [15, 16].

The vast amount of molecular biological data which can be quite easily collected these days (e.g. through genomics or proteomics) provides a rich source of potential predictive biomarkers to define target subgroups. However, from a statistical perspective the high-dimensionality of the resulting problem creates additional challenges compared to the already difficult task of subgroup analysis when only a small number of potential biomarkers is considered. Specifically one is confronted with a high-dimensional variable selection problem to determine the set of biomarkers which actually interact with the treatment.

Depending on the modelling approach a variety of different variable selection methods have been proposed, see for example [17–20]. The most common selection strategies involve different forms of penalized regression for which Lipkovich et al. [3] provide a unified formal framework which allows to include both the parametric and the machine learning approaches. An extremely popular special case is the LASSO [21] which combines the log-likelihood of the regression model with the $L_1$ penalty of coefficients. This was applied by Tian et al. [19] where a modified covariate method was suggested to identify subgroups without having to model the main effects.

Apart from LASSO Lipkovich et al. [3] also mention the use of $L_2$ penalties in ridge regression and the combination of $L_1$ and $L_2$ penalties resulting in the elastic net [22], as well as a number of variations. Among those is the adaptive lasso [23] which turns out to have superior selection and prediction properties. In this article we want to compare the performance of adaptive lasso to identify predictive SNP biomarkers with selection based on FDR controlling $L_0$ penalties and a novel adaptive version of the Sorted L-One Penalized Estimator, also aimed at FDR control. There exists a substantial amount of literature on the theory of $L_0$ penalties which are suitable for model selection in a high-dimensional setting [24, 25]. A comprehensive overview of these methods can be found in two books [26, 27].

We will focus here on modifications of the Bayesian Information Criterion which have been previously successfully applied in the context of QTL mapping [28–30], in genome-wide association studies [31, 32] and for admixture mapping [33]. Specifically we will perform selection based on the mBIC2 criterion, which has been shown to have certain asymptotic optimality properties [34] and allows for control of the False Discovery Rate (FDR) when the predictors are independent or weakly correlated [26, 27]. Furthermore we will make use of SLOBE [35], a Bayesian adaptive version of the Sorted L-One Penalized Estimator (SLOPE). SLOPE [36, 37] is an extension of LASSO obtained by replacing the $L_1$ penalty norm with a Sorted $L_1$ norm, defined through a decreasing sequence of tuning parameters. When the

sequence of these parameters is selected in accordance with the sequence of decaying thresholds of the Benjamini-Hochberg correction for multiple testing [38] then SLOPE adapts to unknown sparsity and achieves an asymptotic minimax rate of estimation and prediction errors under high dimensional sparse linear or logistic regression [39–41]. It also controls FDR when the columns in the design matrix are orthogonal to each other and asymptotically controls FDR when the predictors are independent [42, 43]. Similarly as in case of LASSO, the properties of SLOPE can be further enhanced by using its adaptive version, which weights the tuning parameters according to the expected (estimated) magnitude of regression coefficients. SLOBE, a Bayesian adaptive version of SLOPE, uses weights calculated based on Bayesian principles, similarly as it is done in the Spike and Slab Lasso [44] (the Bayesian version of adaptive LASSO). According to simulation studies reported elsewhere [27, 35], SLOBE has a substantially smaller estimation error and controls FDR for a much wider set of statistical setups than the original SLOPE.

We want to compare the different selection procedures and classical single marker tests (testing each marker individually using marginal tests combined with multiple testing correction) with respect to two different goals. On the one hand we want to identify predictive biomarkers and on the other hand identify those patients which are more likely to benefit from a certain treatment. For both objectives we will compare the performance of SLOBE and mBIC2 based selection with the adaptive lasso. We perform comprehensive simulation studies, where the first set of simulations is based on independent SNPs, whereas the second set uses SNPs which are distrubuted like real genomic data. Finally we will also illustrate the gain in power when testing the effect of treatment both in the selected subgroup as well as in the overall population when applying model selection strategies for genetic data [1–3].

## Materials and methods

For the ease of presentation we will focus here on linear regression but our general ideas can be transferred immediately to other types of regression models. Consider data from $n$ individuals of a two-armed randomized clinical trial with treatment allocation variable $T_i \in \{-1, 1\}$ for each patient $i \in \{1, \ldots, n\}$. The primary outcome $Y_i$ of the study is assumed to be quantitative and additionally we assume to have data $X_{ij}$ from $p$ (genetic) biomarkers (that is $j \in \{1, \ldots, p\}$) for each individual. The model we would like to study will then be of the form

$$Y_i = \mu T_i + \sum_{j=1}^{p} (\beta_j X_{ij} + \gamma_j X_{ij} T_i) + \epsilon_i \,, \tag{1}$$

where $\mu$ encodes the overall treatment effect, $\beta_j$ the prognostic effect and $\gamma_j$ the predictive effect of the $j$-th biomarker, respectively. Again for the sake of simplicity we assume that the individual error terms are Gaussian i.i.d., that is $\epsilon_i \sim \mathcal{N}(0, \sigma^2)$.

### Identifying predictive biomarkers

We are mainly concerned with a high-dimensional setting, for example when SNPs are considered as potential biomarkers. In case of $p$ SNPs the number of prognostic and predictive biomarkers is $2p$, which is potentially much larger than the sample size $n$. Then the coefficients of the full model (1) are no longer estimable and some kind of regularization or model selection must be applied before identifying predictive biomarkers.

We consider different model selection approaches where we only select on biomarkers but not on the treatment effect $\mu$. The simplest method consists of testing both prognostic and predictive bio-marker coefficients individually and adjust for multiplicity using either Bonferroni

or Benjamini-Hochberg corrections, respectively. A more advanced model selection strategy is based on minimizing the mBIC2 criterion [31]

$$\text{mBIC2} = n \log \text{RSS} + k \log n + 2k \log(2p/4) - 2 \log(k!) \quad, \tag{2}$$

where $k$ is the number of biomarker coefficients in a specific model and RSS is the corresponding residual sum of squares from classical least squares regression. This criterion was designed to control the FDR of wrongly detected regressors under sparsity when both $n$ and $p$ are large and the predictors are independent or weakly correlated. To identify the model with the optimal value of mBIC2 we use an advanced and efficient step-wise selection procedure implemented in the *R* package *bigstep* [45]. According to the simulations reported e.g. in [33] or [27] this search strategy allows to identify models very close to the optimal one when predictors are roughly independent, like in the case of Genome Wide Association Studies.

As a second advanced model selection approach we consider SLOBE [35], a Bayesian adaptive version of SLOPE [36, 37]. Minimizing mBIC2 is a difficult mixed integer program, whereas SLOPE is performed by solving the following convex optimization program

$$\min_{b \in \mathbb{R}^{2p}} \| Y - X^{tot} \|_2^2 + \sum_{j=1}^{2p} \lambda_j |b|_{(j)} \quad, \tag{3}$$

where $X^{tot}$ is the design matrix including both prognostic and predictive biomarkers and $b = (\beta_1, \ldots, \beta_p, \gamma_1, \ldots, \gamma_p)^T$ is the vector of all biomarker coefficients. The notation $|b|_{(j)}$ indicates that the coefficients are ordered according to their absolute value. When the design matrix is standardized such that each column has a unit $L_2$ norm, the weights $\lambda_j = \sigma \Phi^{-1} \left( 1 - \frac{jq}{2p} \right)$ corresponding to the sequence of decaying Benjamini—Hochberg thresholds guarantee the control of FDR when the design matrix is orthogonal and asymptotic FDR control if the predictors are independent.

SLOPE is computationally much less intensive than mBIC2, but it suffers from problems related to the excessive shrinkage of large regression coefficients. In [46] it is explained in the context of LASSO, that the excessive shrinkage leads not only to inferior predictive performance but also to increased variance of regression estimates and substantial problems with identification of true regressors. These side-effects can be substantially reduced if large regression coefficients are "debiased" using smaller values of the tuning parameters. This idea was used in adaptive LASSO and its Bayesian version, Spike and Slab Lasso, and it is also used for SLOBE, which relies on many iterations of the weighted SLOPE procedure

$$\min_{b \in \mathbb{R}^{2p}} \| Y - X^{tot} \|_2^2 + \sum_{j=1}^{2p} w_j \lambda_{r(b,j)} |b_j| \quad, \tag{4}$$

where $r(b, j) \in \{1, 2, \cdots, 2p\}$ is the rank of $|b_j|$ among absolute values of the coordinates of $b$ in a descending order. In SLOBE the weight $w_j$ depends on the posterior probability that $X_j^{tot}$ is a true predictor and is based on the estimator of $b_j$ from the previous steps as well as on the current estimator of the overall signal sparsity and its average strength. The weight function is designed such that very small regression coefficients are penalized according to the original SLOPE penalty, which allows for control of the number of false positives, while the smaller penalty for large coefficients results in reduction of the estimation bias. According to simulation results reported elsewhere [27, 35], SLOBE has substantially reduced prediction error and improved FDR control compared to the original SLOPE. In our simulation studies we used SLOBE with the parameter $q = 0.05$ chosen to control FDR at a level of 5%. Additionally, due

to the large computational complexity of SLOBE, in case when $p \geq 1000$ we performed initial screening of variables, keeping only 500 columns in the design matrix with the largest correlation with the response variable. To correct for multiple testing, SLOBE on the reduced data is performed at the nominal FDR level $q = 0.05 \times \frac{500}{2p}$.

Additionally, we considered the adaptive LASSO [23] which serves as the benchmark method for building predictive regression models. For simulations we used the R package *adalasso*. The corresponding model of the two-stage adaptive LASSO [47] takes the following form in our context:

$$\min_{b \in \mathbb{R}^{2p}} \| Y - X^{tot} \|_2^2 + \lambda \sum_{j=1}^{2p} \hat{\omega}_j |b_j| \quad . \tag{5}$$

The weights $\hat{\omega}_j$ are defined as the inverse of the coefficients from a regular LASSO search, $\hat{\omega}_j = 1/|\hat{b}_{j,\text{lasso}}|$. Both for the initial LASSO and for the second stage weighted LASSO the penalty parameters $\lambda$ are obtained via crossvalidation.

Thus we study three different methods to select the set of prognostic markers $I_\beta$ and the set of predictive markers $I_\gamma$. In each case the resulting predictive model (given $I_\beta$ and $I_\gamma$) is of the form

$$Y_i = \mu T_i + \sum_{j \in I_\beta} \beta_j X_{ij} + \sum_{j \in I_\gamma} \gamma_j X_{ij} T_i + \epsilon_i \;, \tag{6}$$

where both the sets $I_\beta$ and $I_\gamma$ as well as the coefficients $\mu$, $\beta_j : j \in I_\beta$ and $\gamma_j : j \in I_\gamma$ have to be estimated. In terms of biomarker identification we are interested both in the power to detect relevant predictive biomarkers, prognostic biomarkers and the corresponding false discovery rates of different selection methods.

## Subgroup selection and testing of treatment efficacy

Treatment efficacy means that under the highly controlled circumstances of a clinical trial patients receiving the treatment have a better outcome than patients from a control group. Researchers are often interested whether a new treatment is efficacious in their full study population. If this is not the case then they might still be interested whether there exists any subgroup of patients for which the treatment is efficacious. In the following we want to consider strategies to test efficacy in the overall population or efficacy in a subgroup.

The first three strategies are designed to test efficacy in the overall population using the full data sample. The next method is concerned with efficacy in a subgroup. After splitting the sample in a training and a test data set, a model is estimated in the training data set based on which the so called predictive index (defined in the next paragraph) is computed in the test data set. The predictive index allows to define a subgroup of potential responders in which efficacy can be tested. From the perspective of the researcher it would be ideal to have a strategy which is powerful in both situations. Thus the last method will combine strategies for testing overall efficacy and efficacy in a subgroup.

Model (6) with index sets depending on the selection method can be applied to calculate the patient's predictive index

$$R(X_{i.}) = E(Y_i|X_{i.}, T_i = 1) - E(Y_i|X_{i.}, T_i = -1) \;. \tag{7}$$

Apparently given $I_\gamma$ it holds that $R(X_{i.}) = 2(\mu + \sum_{j \in I_\gamma} \gamma_j X_{ij})$. This index may be used to identify groups of patients for personalized therapies, where a patient with a positive predictive index

$R(X_{i.}) > 0$ is defined to be responsive to the treatment. Estimates of the predictive index obtained with different methods are generically denoted as $\hat{R}(X_{i.}) = 2(\hat{\mu} + \sum_{j \in \hat{I}_\gamma} \hat{\gamma}_j X_{ij})$.

The following list describes the five strategies to test for efficacy in the overall population or in a subgroup in more detail. Although these strategies are testing different hypotheses it is of some interest to compare their power to detect efficacy in various scenarios because they reflect the situation which researchers actually have to face. A priori they do not know whether a new treatment works for the full population or only for a subgroup. To keep things tractable we are using for this part only the mBIC2 method to select (prognostic and predictive) biomarkers.

- **Test whether there is an overall treatment effect in the full study population**: The first three strategies are using all available samples and test at a significance level $\alpha = 0.05$.

  - **Method 1**: The simplest approach is to entirely ignore biomarkers and only apply a t-test for the treatment effect.

  - **Method 2**: Test the coefficient $\mu$ in a regression model (6), where prognostic and predictive markers are selected with mBIC2.

  - **Method 2a**: As before test the coefficient $\mu$ in a regression model (6) selected with mBIC2, but including only prognostic markers. In contrast to Method 2 the influence of the treatment is represented by $\mu$ only and interaction effects are ignored.

- **Test whether there is a treatment effect in a biomarker defined subgroup**: The next strategy is based on splitting the sample. The first half of patients is used to select prognostic and predictive markers with model (6). A model based on these markers is subsequently used to select those patients for whom $\hat{R}(X_i) > 0$.

  - **Method 3**: Apply a simple t-test for the treatment effect (like in Method 1) but only for responders in the test data set at a significance level $\alpha = 0.05$.

- **Test whether there is a treatment effect overall or in a biomarker defined subset**: The final strategy combines a test for an overall treatment effect with the test for efficacy in a subgroup. The first test is performed using all patients at half of the nominal alpha level. The second test, again performed at half of the nominal alpha level, uses only predicted responders from the second group, where as before, the responders in the second group are estimated using the model selected and fitted on the patients from the first group. A treatment is identified as effective if at least one of these tests rejects the null hypothesis of no treatment effect.

  - **Method 4**: Combination of tests from Method 2 and Method 3 at a Bonferroni corrected significance level $\alpha = 0.025$.

**Remark**: In principle Method 3 could be improved by performing a model based test which includes only prognostic markers (like in Method 2a). However, in our simulations the gain of power was rather neglectible in this case and therefore we do not present results for this method.

## Simulations

**Part 1**. In our first simulation study we want to compare the performance of different selection methods to correctly identify predictive biomarkers and consequently identify sets of responsive patients. To this end we simulate randomized clinical trials with $n = 1000$ patients and two different dimensions of the genotype data, which consisted of $p = 100$ and $p = 2000$ distant

(independent) Single Nucleotide Polymorphisms (SNPs), respectively. The SNP genotypes were simulated assuming Hardy-Weinberg equilibrium. Minor allele frequencies were randomly selected from the interval (0.1, 0.5), separately for every SNP. We simulated several scenarios, with the number of causal SNPs influencing the response, $k$, belonging to the set {2, 6, 10, 30, 50}. In each scenario half of the causal variants were prognostic and the other half predictive. The main treatment effect $\mu$ was set to be zero and the regression coefficients for the interaction effects were positive or negative with the same probability, which resulted in approximately 50% of patients responding positively to the therapy ($R(X) > 0$).

Before doing simulations the genotype matrix was centered and scaled, so that each column has zero mean and a unit $L_2$ norm. One consequence of this scaling is that the power of identifying markers with a given effect size $\beta$ does not depend on the sample size $n$. Trait values were then simulated by setting the absolute values of all nonzero regression coefficients to be equal to $1.5\sqrt{2\log p}$ and choosing a positive or negative sign with equal probability. The simulation scenarios are not completely realistic from a genetic perspective because one would not expect that all markers have the same effect size. However, they perfectly illustrate the behaviour of the different selection methods. The second simulation study described below will then illustrate the performance of different methods using markers which are distributed like SNPs from real genomic data.

For the case $n = 1000$ and $p = 100$ we additionally used the classical least-square approach for fitting the predictive model (1) based on all available markers (including non-important variants). The characteristics of statistical procedures were obtained by averaging results from 1000 independent simulations. In this way we estimated for each procedure the power and the false discovery rate (FDR) to detect prognostic and predictive markers, the standardized mean squared error of the predictive index

$$MSE = \frac{\sum_i (R(X_{i.}) - \hat{R}(X_{i.}))^2}{\sum_i R^2(X_{i.})} \; ,$$

and finally the percentage of correctly reported treatment responders (positive detection rate) and the percentage of wrongly reported non-responders (negative detection rate).

**Part 2**. The second simulation study is similar to the first one but is based on correlated SNPs as potential biomarkers. Genotype data stem from an admixture population and were generated for $n = 1000$ individuals as described by Szulc et al. [33]. To keep the computational effort for simulations within reasonable limits we have used only the genotype data of $p = 7297$ SNPs from one chromosome and simulation results are based here on 500 replications. Due to linkage disequilibrium (LD) the correlation structure of SNPs is typically a block structure where neighboring SNPs are often strongly correlated and form clusters. Selection procedures might find it difficult to distinguish between highly correlated SNPs and therefore one might consider all SNPs detected within the same cluster as the causal SNP itself as true positive findings. For this purpose we clustered SNPs with the *clump_snps* function from the R-package *geneSLOPE* which is available at https://github.com/psobczyk/geneSLOPE. The genotype data and the clustering information are provided in the file *SNP_data.RData*.

The second simulation study has several aims. Apart from looking at the effect of correlation on the performance of the different selection methods we want to answer the question whether it makes a difference for a marker to be both prognostic and predictive at the same time, or to be only prognostic or predictive. Furthermore we want to study to which extent the power to detect a specific marker depends on the size of the cluster in which it is located. For this reason we consider a simulation scenario where 15 SNPs are specified as regressors in the data generating model. 5 of them are purely prognostic, 5 are purely predictive and 5 are both

prognostic and predictive. Within each of these three sets the 5 SNPs are chosen to be members of a cluster of size 1, 2, 3, 5 and larger than 6, respectively. The 15 SNPs used for the data generating model are almost uncorrelated with a maximal pairwise correlation of 0.17. More details on the genotype data and on the SNPs used for the data generating model can be found in S3 Appendix.

Selection based on single marker tests (both Bonferroni and Benjamini Hochberg) as well as selection based on mBIC2 involves penalties which directly depend on the total number of SNPs. In case of correlated SNPs the resulting correction for multiplicity tends to become too severe and one might prefer to use a somewhat smaller penalty. One possibility is to make use of an 'effective number of markers', a concept which has been described for example Bogdan et al. [29] in the context of QTL mapping. The general idea is to simulate the distribution of the maximal test statistic over all markers under the null hypothesis and thereby estimate the critical value of this maximum statistic which corresponds to a desired significance level. On the other hand it is easy to compute the critical value of the maximum of independent test statistics (see equation (4) of [29]). The effective number of markers is then defined as the number of independent tests which gives the same critical value as the one obtained from simulations for the correlated markers. For our SNP data the efficient number obtained from simulations was $p_{eff} = 13700$, compared to the total number of $2p = 14594$ prognostic and predictive markers. We substituted $2p$ by $p_{eff}$ for the Bonferroni correction, for the Benjamini Hochberg procedure and for the penalty term of the mBIC2 criterion.

Like in the first simulation the genotypes of all SNPs were standardized and half of the individuals were randomly allocated to the treatment group. Effect sizes were again chosen to be equal for all markers with effect size $\pm c_{eff} \sqrt{2 \log p}$. Five different scenarios were considered with $c_{eff} \in \{1.2, 1.3, 1.4, 1.5, 1.6\}$. The performance of the different selection procedures was assesed as described above but additionally we considered the power to detect individual SNPs. To account for the specific correlation structure of SNPs described above two different definitions of true positive detections were introduced. First we declared a SNP entering a selected model as true positive only when it coincided exactly with the SNPs from the data generating model. Secondly we considered a more relaxed definition when a SNP which was within the same cluster as a SNP from the data generating model was also declared as a true positive finding.

**Part 3**. In the final set of simulations we evaluated the five proposed methods for testing treatment efficacy in four different scenarios. These are described in Table 1. Scenario 1 is more or less identical to the simulation setting from Part 1, except that the total number of SNPs was $p = 500$. For Scenario 1 the overall treatment effect was $\mu = 0$. Scenario 2 had $\mu = 0.07$ but was otherwise identical with Scenario 1. For Scenario 3 with $\mu = 0.1$ and Scenario 4 with $\mu = 0.12$ the treatment effect was further increased, while the effect size of prognostic and predictive markers was decreased with $c_{eff} = 1.3$ in Scenario 3 and $c_{eff} = 1.2$ in Scenario 4, compared with $c_{eff} = 1.5$ in the first two scenarios.

The percentage of variability of the trait explained by genetic factors is called heritability. S1 Appendix provides the details how to compute the heritability for our simulation scenarios as presented in Table 1. To assess the performance of the five different methods we compare the power to detect treatment efficacy either in the whole population or at least in a subgroup. For the first three methods the efficacy in the full population is tested, for Method 3 we test whether the treatment is effective within a subgroup and the last method combines the questions whether the treatment is effective in the whole population or only in a subgroup. Although these methods are testing different hypotheses it is of interest to compare the power of the different strategies to be successful in the sense that they identify efficacy either in the

**Table 1. Description of simulation scenarios.**

| Sc | $\mu$ | $c_{eff}$ | k = 2 | k = 6 | k = 10 | k = 30 | k = 50 |
|----|-------|-----------|-------|-------|--------|--------|--------|
| 1 | 0 | 1.5 | 5.3 (0.0) | 14.4 (0.0) | 21.9 (0.0) | 45.6 (0.0) | 58.3 (0.0) |
| 2 | 0.07 | 1.5 | 5.3 (0.5) | 14.3 (0.4) | 21.8 (0.4) | 45.5 (0.3) | 58.2 (0.2) |
| 3 | 0.1 | 1.3 | 4.0 (1.0) | 11.1 (0.9) | 17.2 (0.8) | 38.4 (0.6) | 51.0 (0.5) |
| 4 | 0.12 | 1.2 | 3.4 (1.4) | 9.6 (1.3) | 15.0 (1.2) | 34.6 (0.9) | 46.9 (0.8) |

All four scenarios (Sc) for the second set of simulations had $n = 1000$ and $p = 500$. $\mu$ refers to the overall treatment effect. Non-zero coefficients of genetic markers were set to $c_{eff}\sqrt{2\log p}$. The last five columns provide the heritability (in percentages) for the genetic effects and the treatment (in brackets) for different $k$.

complete population or in a subgroup. Power estimates are based here on 1000 simulation replicates.

## Results

### Simulation Part 1: Biomarker identification and subgroup selection

Fig 1 illustrates the performance of the different selection procedures with respect to correctly identifying biomarkers. In these plots we do not distinguish between prognostic and predictive markers as we are mainly concerned with the general accuracy of methods to detects markers. Detailed results distinguishing between prognostic and predictive markers are provided in S2 Appendix. However, in this simulation study based on independent markers there is hardly any difference in detection rates between prognostic and predictive biomarkers.

Clearly the model selection methods SLOBE, mBIC2 and adaptive LASSO have much larger power than methods based on single marker tests when the number of causal variants, $k$, is moderate or large, particularly in the high-dimensional case. Single marker tests are losing power with increasing complexity of the data generating model because in the denominator of the t-test statistic the variance is overestimated when the effect of a large number of causal variants is not taken into account (see [31] for more details).

While the adaptive LASSO has the largest power in all scenarios it has also the largest type I error rate by far. SLOBE has slightly larger power than mBIC2 for $p = 100$ and almost identical power for $p = 2000$. In both scenarios mBIC2 is controlling the FDR at a level slightly below $\alpha = 0.05$. SLOBE controls FDR right at $\alpha = 0.05$ in case of $p = 100$ as well as for $p = 1000$ and small $k$. For more complex models in the high-dimensional setting SLOBE suffers from some inflation of the type 1 error, which is mainly due to the inaccuracy of the correlation ranking (see e.g. [31]) and the related imprecision of the preselection procedure. However, even in this range FDR of SLOBE remains smaller than that of the adaptive LASSO, which has FDR ranging between 0.25 and 0.31 for both scenarios. Note that the Bonferroni correction is extremely conservative whereas the Benjamini Hochberg procedure keeps the FDR at the desired level of $\alpha = 0.05$ similar to mBIC2.

To summarize, the model selection procedures are in general much better suited for model identification than the marginal tests. mBIC2 has only slightly smaller power than the adaptive LASSO but a much smaller type I error rate. The same is true for SLOBE which only has a slightly inflated type I error rate for more complex models in the high-dimensional scenario.

Fig 2 is concerned with how well different procedures can estimate the predictive index $R$ $(X)$. Based on the results on biomarker identification it is not too surprising that with increasing number of causal variants the procedures based on single marker tests perform worse than the model selection procedures. For the low-dimensional scenario the precisions of adaptive

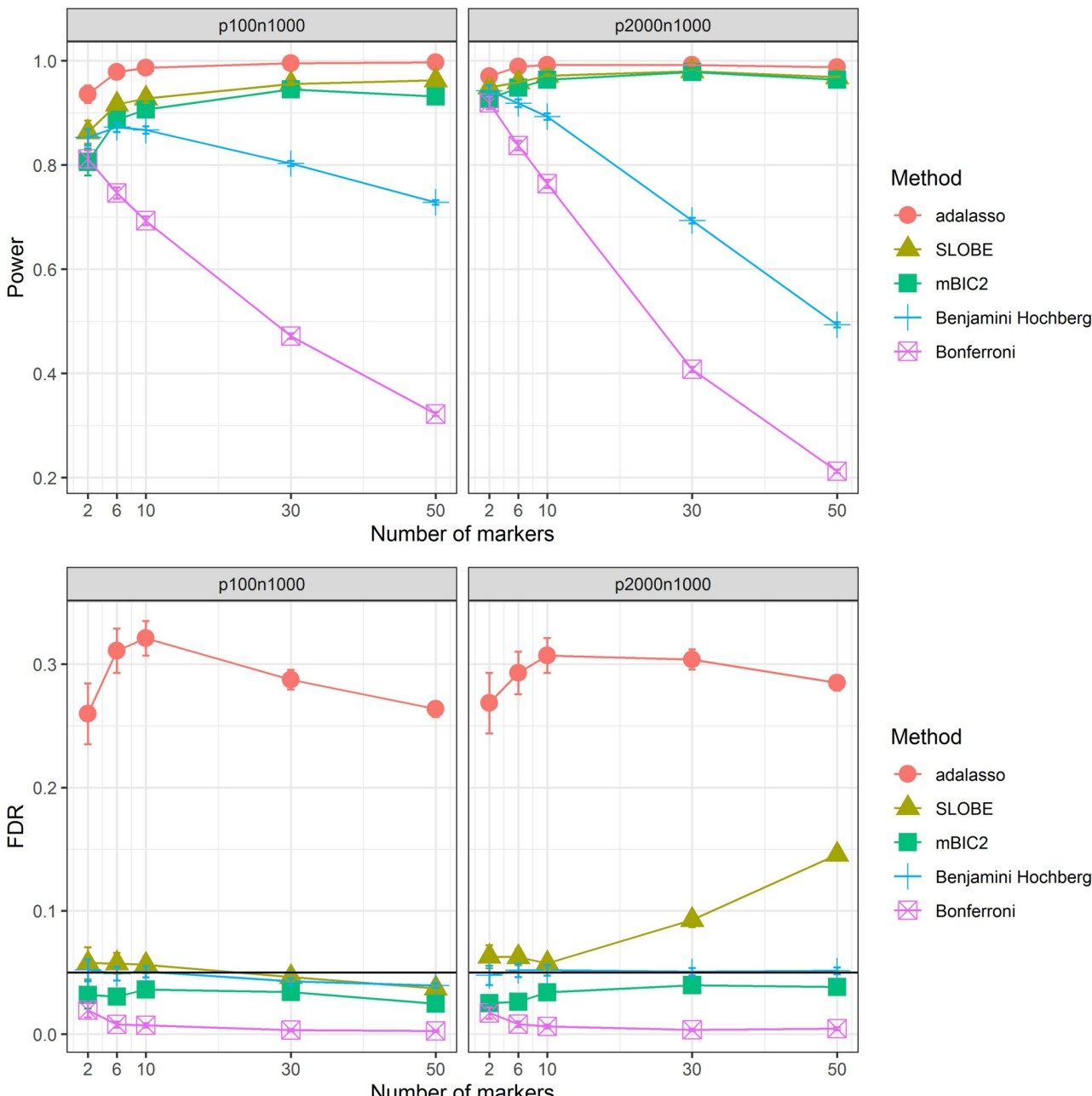

**Fig 1. Biomarker identification.** Power and False Discovery Rate of biomarker identification as a function of the number of causal variants $k$ for two different simulation scenarios ($p < n$ and $p > n$).

lasso, mBIC2 and SLOBE are very similar, with mBIC2 having slightly larger prediction error for the most complex model. In the high dimensional scenario mBIC2 outperforms the adaptive LASSO for the whole range of model complexities. SLOBE performs very similarly to mBIC2 for the small models, while its mean squared error gets closer to the adaptive LASSO for the more complex models. Here it is worth to remember that mBIC2 and SLOBE are using substantially less SNPs for the predictive models than adaptive LASSO and are still achieving comparable or even better predictive quality.

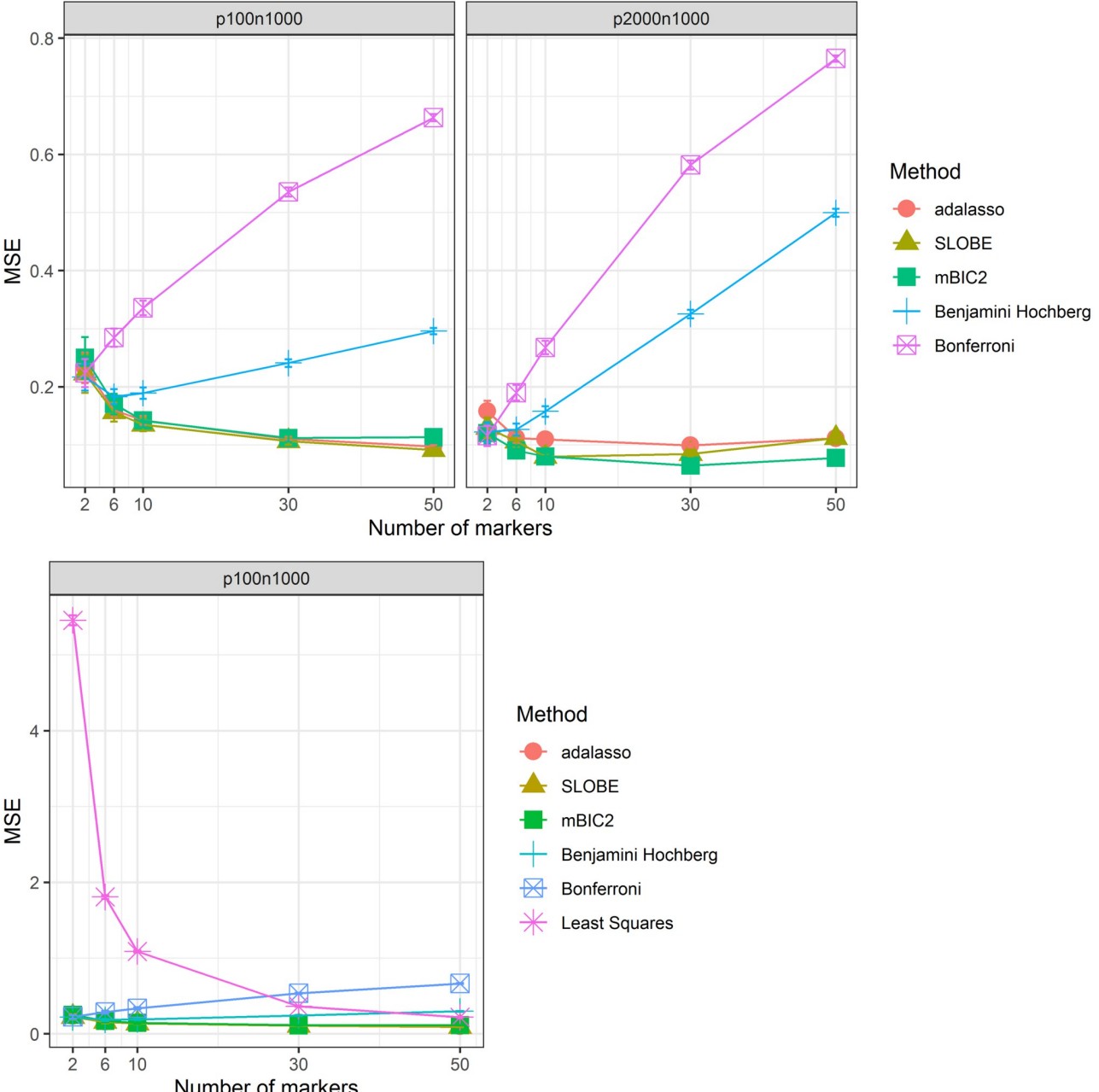

**Fig 2. Estimation of predictive index.** Mean Squared Error of the estimated predictive index $R(X)$ as a function of the number of causal variants $k$ for two different simulation scenarios ($p < n$ and $p > n$). The last panel includes MSE for the ordinary least squares estimate (LSE).

In case when $p = 100$ and $n = 1000$ one could in principle omit the selection step and estimate regression coefficients using all biomarkers. However, the last panel of Fig 2 shows that this leads to a rather disastrous prediction if in reality only very few genes influence the trait (say $k \leq 10$). This phenomenon is due to the relatively large variance of estimates of regression coefficients one obtains for $p = 100$. Model selection methods allow to reduce the number of predictors and the related variance of regression estimates.

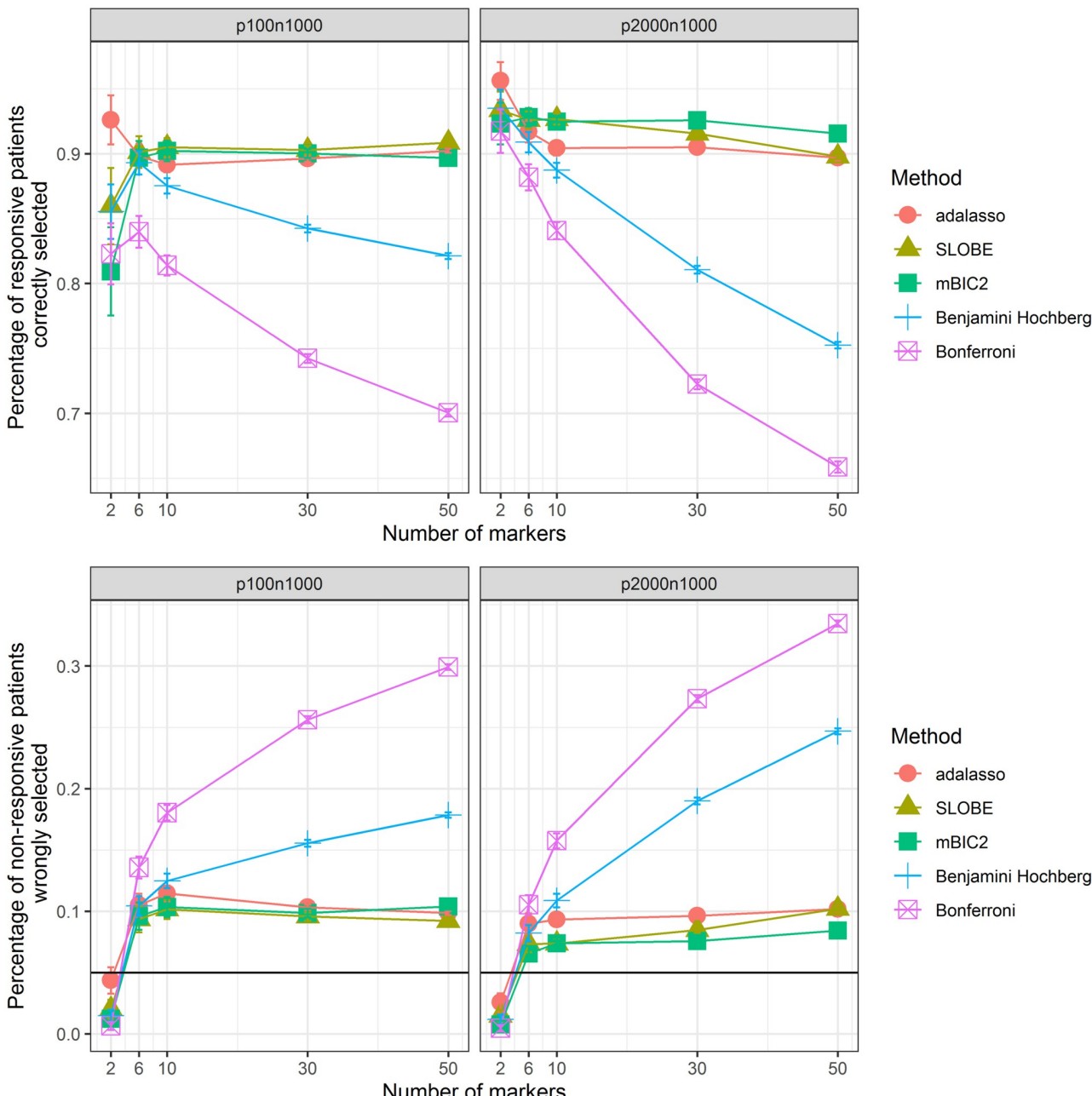

**Fig 3. Identification of responders and non-responders.** First the percentage of responsive patients (true $R(X) > 0$) identified by different methods (estimated $R(X) > 0$) as a function of the number of causal variants $k$ for two different simulation scenarios ($p < n$ and $p > n$). Then the percentage of non-responsive patients (true $R(X) < 0$) wrongly selected by different methods (estimated $R(X) > 0$).

Fig 3 is concerned with the correct detection of responders. Specifically, the plots from the first row provide the percentage of responsive patients which were identified by different methods, while plots from the second row give the percentage of non-responsive patients in the group where $\hat{R}(X) > 0$. Here again we can see the superior performance of the model selection methods. In the low dimensional setup mBIC2, SLOBE and adaptive LASSO perform

fairly similar, with SLOBE slightly outperforming other methods for more complex models. In the high dimensional scenario mBIC2 is the best (with the exception of $k = 2$), with SLOBE being equally good for $k < 30$.

The adaptive lasso as benchmark allows to identify around 90% of responsive patients at an FDR level of around 10% (except for $k = 2$). Methods based on multiple testing have a substantially lower sensitivity and at the same time an undesirably high FDR level. For example, in case of the regression model created with the help of the Bonferroni correction, for $k = 50$ more than 30% of the predicted responders ($\hat{R}(X) > 0$) have in reality a negative predictive index. This clearly illustrates that if one expects a larger number of predictive biomarkers then one should not try to identify them using marginal tests but instead rely upon model selection strategies.

## Simulation Part 2: Real SNP data

We first want to discuss the results for correlated SNPs based on the strict definition of true positive detections, where only those detections are considered as True Positives which conincide with the SNPs from the data generating model. S2 Appendix provides the figures for the overall analysis which are analogous to Figs 1–3 from the first set of simulations. Here the power is taken as the average power over the 20 causal markers from the data generating model. Concerning the model selection procedures the general behaviour is qualitatively rather similar to the case of independent markers. Adaptive LASSO has the largest power but also the largest FDR, SLOBE and mBIC2 have quite similar power but SLOBE has a slightly larger FDR than mBIC2. The Bonferroni procedure also behaves as expected, controlling the type I error rate very strictly but being the least powerful procedure.

The biggest difference we observe lies in the behaviour of the Benjamini Hochberg procedure, which has now a power in the range of the model selection procedures but at the expense of a hugely inflated type I error. This can be explained as follows. With each positive detection the Benjamini Hochberg procedure increases the level with which the sorted p-values are compared. Thus it becomes easier for markers to become selected by the procedure when there have been selected already many other markers. In our setting most SNPs from the data generating model are strongly correlated with other SNPs. These correlated SNPs will all have a fairly large chance to be detected. As a consequence the probability for detecting further false positives which are not correlated with causal SNPs is severely increased. This is apparently a well known problem when applying the Benjamini Hochberg procedure to analyse GWAS data and was discussed in detail by [48] who also introduced a remedy for this issue. Therefore we do not want to go into more detail here but just remark that a naive application of the Benjamini Hochberg procedure in case of correlated markers has its pitfalls.

We now want to discuss in more detail the detection rates for individual SNPs from the data generating model. Fig 4 presents the corresponding results for an effect size of $1.5\sqrt{\log p}$. The results for other effect sizes are provided as supplementary material in S2 Appendix. Among the model selection procedures adaptive LASSO almost universally has the largest power to detect individual SNPs. mBIC2 tends to have slightly larger power than SLOBE for most SNPs but there are some exceptions.

The first plot of Fig 4 shows that there are five SNPs where the methods based on model selection have substantially lower power than for other SNPs. For these specific predictors they have a tendency to include correlated SNPs rather than the causal SNP from the data generating model. Two of these SNPs are purely predictive, one is purely prognostic and two are both predictive and prognostic. The purely predictive markers SNP 1484 and SNP 5902 belong to clusters of size 5 and 8, respectively. The prognostic marker SNP 1888 belongs to a cluster of

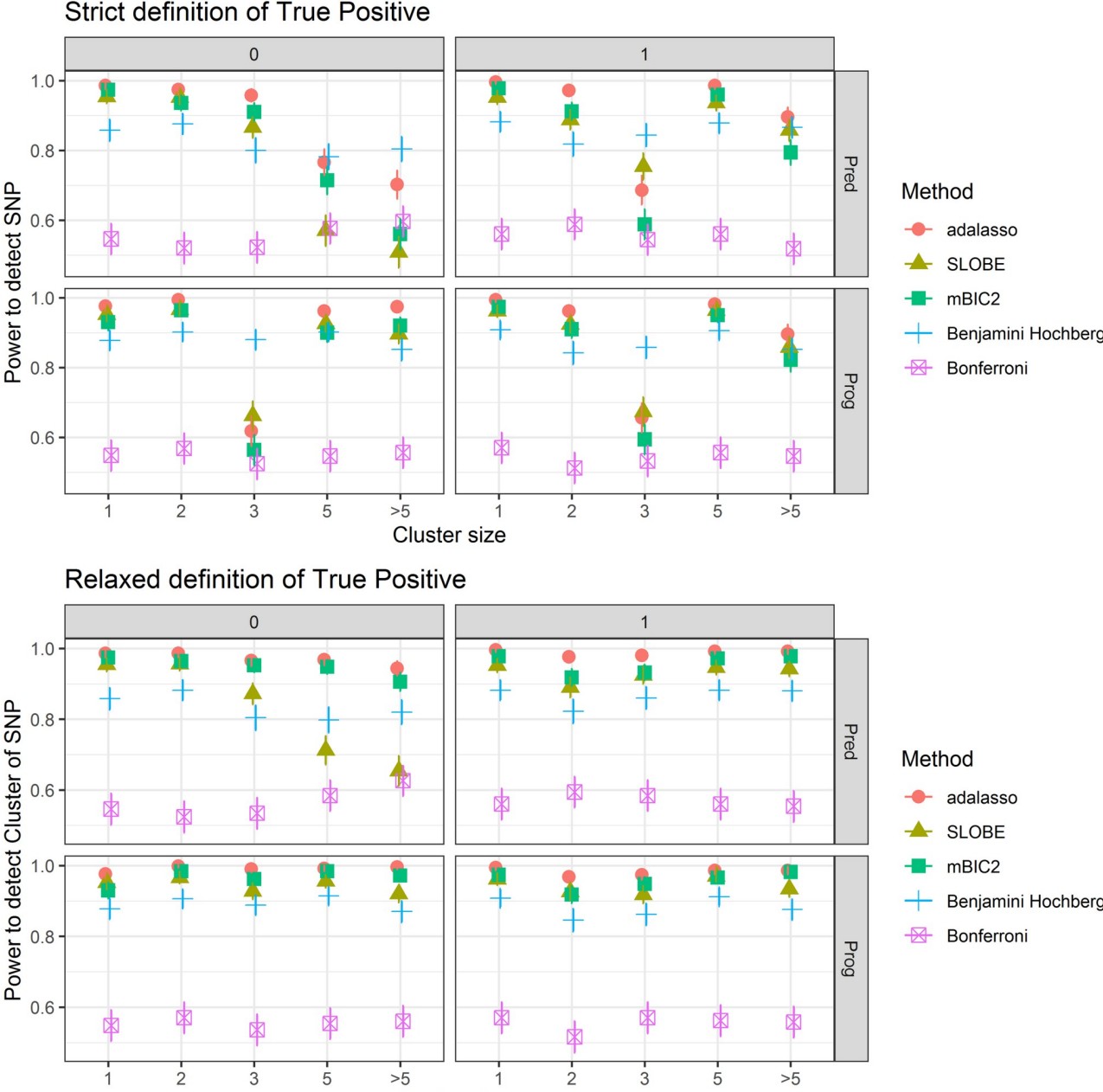

**Fig 4. Power to detect individual SNPs.** SNPs are arranged according to three different criteria. The rows of the facets indicate whether SNPs are predictive (first row) or prognostic (second row), The columns of the factets indicate whether they are specifically of one type (left column) or both predictive and prognostic (right column). Within each facet SNPs which belong to larger clusters are found further to the right. The first plot is based on a strict definition of true positives where the detected SNP must coincide with the SNP from the data generating model. The second plot uses the more relaxed definition where a detected SNP is counted as true positive whenever it is within the same cluster as a SNP from the data generating model.

size 3 and the markers SNP 4052 and SNP 6619, which are both predictive and prognostic, belong to clusters of size 5 and 7, respectively. So it seems that this phenomenon depends more on the specific correlation structure within a cluster belonging to a particular SNP and not so much on whether a marker is purely predictive, purely prognostic or both.

The second plot of Fig 4 presents the results for the relaxed definition of true positives, where all SNPs within the same cluster as the SNP from the data generating model are considered as true positives. As expected there is essentially no difference in power for the methods based on single marker tests. Note that using the cluster based definition the Benjamini Hochberg procedure is actually less powerful than the three model selection based methods in spite of its large type I error rate. Both for adaptive LASSO and for mBIC2 the power becomes universally large for all SNPs at an effect size of $1.5\sqrt{\log p}$, whereas SLOBE still has lower power for the three purely predictive markers which belong to clusters of size 3, 5 and 8, respectively.

Concerning the estimation of the predictive index and also the classification of responders mBIC2 performs best for larger effect sizes whereas the adaptive LASSO outperforms the other methods for the smallest effect size. SLOBE consistently performs worse here in terms of classifying responders correctly. The two methods based on marginal tests perform even worse. One should also mention that for the fairly sparse models of this simulation scenario the stepwise search for mBIC2 is much faster than both SLOBE and adaptive LASSO. Whereas it took *bigstep* only a few hours to run all the simulation scenarios for the SNP data, SLOBE needed almost a day and *adalasso* needed several days. Here the main bottleneck is the cross-validation of LASSO, which is extensively used in *adalasso* and is also used by SLOBE for estimating the standard deviation of the error term. Apparently the runtime of mBIC2 will substantially increase for data generating models which include more causal SNPs whereas for cross-validated LASSO the runtime should not depend too much on model complexity.

## Simulation Part 3: Testing of treatment efficacy

Fig 5 illustrates the results from the third simulation study. In the first scenario the marginal treatment effect is equal to zero and therefore both the regular t-test and the F-tests using regression models selected by mBIC2 (red lines) have no chance to detect treatment efficacy. Their power is more or less at the nominal type I error level of 0.05. The strategies based on sample splitting tend to perform rather well for larger numbers of causal variants. With increasing heritability the detection of predictive markers in the training sample gives better prediction of responders in the test data set. The power to identify the efficacy of the test in the sub-group of predicted responders ranges from almost 0% when there is only one predictive marker to almost 100% when there exist many genes interacting with the treatment.

We want to emphasize that in this simulation study the marker effects are rather weak which makes these scenarios fairly challenging. With the full data set the power to detect causal markers is around 90% (depending on $k$) wheres after sample splitting the power drops considerably (between 35% and 45%). Consequently also the power to predict responders correctly in the test sample is only around 70% going along with a false discovery rate from up to 30%. However, even this fairly small prediction accuracy of responders is sufficient to successfully show that the treatment is effective in a subgroup for larger $k$. For small $k$ there are many simulation runs where no predictive marker is detected at all and in that case one cannot determine a subgroup in which to test for efficacy. This is the reason why particularly for $k = 2$ the power obtained with Method 3 is close to zero.

The first scenario is the only one where the strategy of Method 3 to test within the subgroup of responders (blue lines in Fig 5) performs better than the combination test of Method 4 (black lines in Fig 5). This is a direct consequence of the fact that for $\mu = 0$ tests on the whole population do not provide any information on efficacy.

Scenario 2 is almost the same as Scenario 1, but now there is also a small marginal treatment effect which is on the "verge of detectability". This means that the power of the marginal test within the regression model selected by mBIC2 is close to 60% under all considered genetic

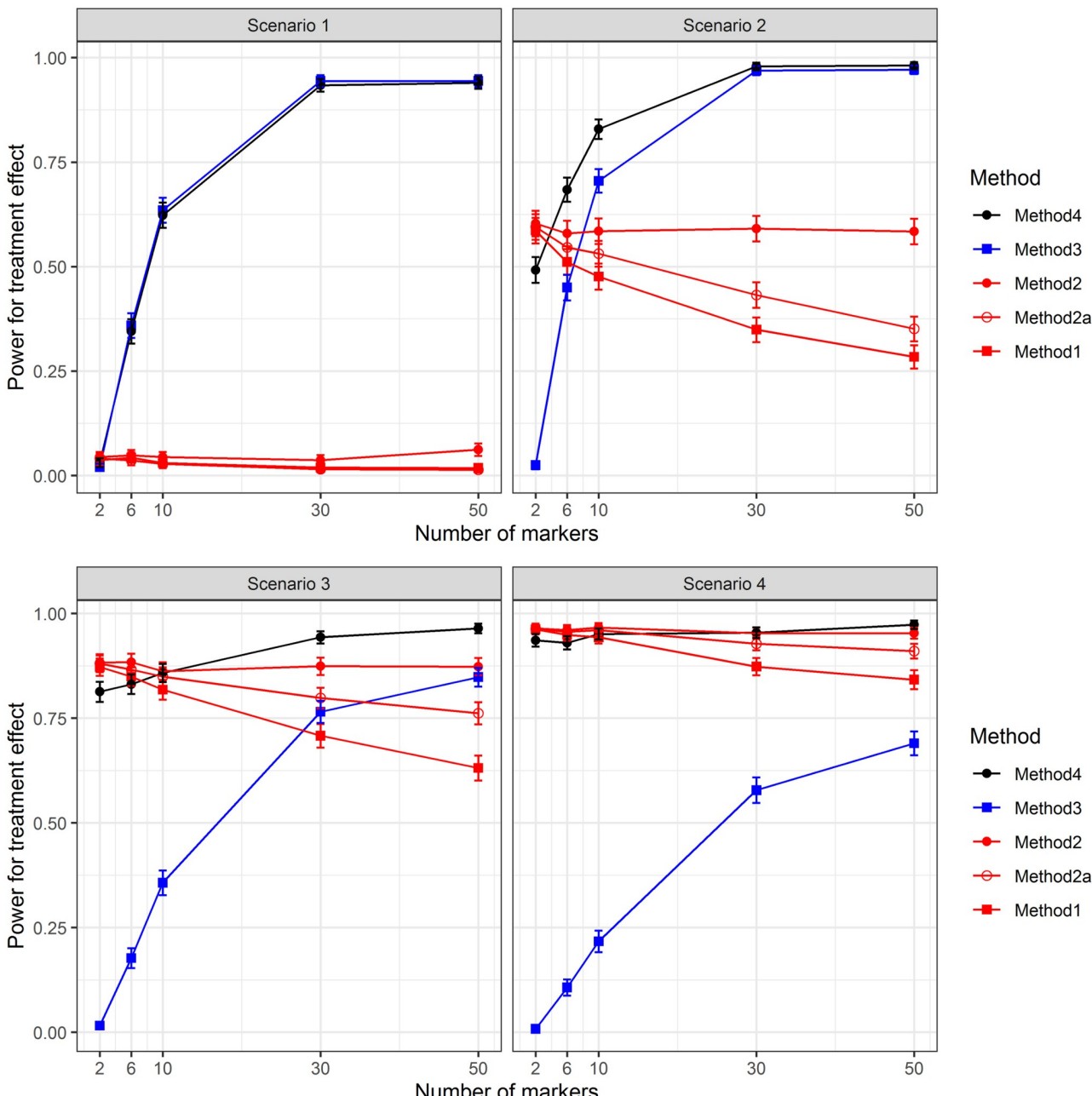

**Fig 5. Treatment efficacy.** Power to detect treatment efficacy for the four scenarios from the second simulation study.

scenarios. The second panel of Fig 5 illustrates that Method 1 suffers from a large loss of power compared with Method 2. The reason for this is the same as we discussed already in the first simulation study. A two-sample t-test where the genetic effects are not taken into account results in strong inflation of the residual variance. Specifically, in case when there are 50 genes influencing the trait the power of the regular t-test drops to 20% as compared to 60% provided by the test using the regression model determined by mBIC2. Including only prognostic markers in the model (Method 2a) increases the power compared to simple t-tests of Method 1 but

the procedure is not as powerful as including both prognostic and predictive markers in the model. When including only prognostic markers the estimate of the residual variance is still including the contribution of the predictive markers.

In this scenario again Method 3 based on sample splitting performs rather well for larger $k$. However, only for $k > 6$ it becomes better than tests based on the whole data set (red lines). In particular for $k = 6$ even marginal tests are slightly more powerful than Method 3. However, the combination of the two strategies proves very useful both for small $k$ where the marginal treatment effect gives valuable information on efficacy and for large $k$ where the interaction effects between treatment and genetic markers allow to detect the subgroup for which the treatment is particularly effective. Except for $k = 2$ the combination test is most powerful.

In Scenario 3 and Scenario 4 the marginal treatment effect is further increased whereas the marker effects are decreased. Consequently marginal tests using the whole data set (red lines) tend to perform increasingly well. Method 2 has a power of more than 85% for Scenario 3 and more than 95% for Scenario 4 irrespective of the number of causal markers. Once again the marginal t-test for the treatment effect performs much worse. While Method 3 which is restricted to the selected group of patients in the test data set (blue line) performs much worse than Method 2, the combination test of Method 4 (black line) is not much worse than Method 2 for small $k$ and is performing best for larger numbers of causal markers.

## Discussion

The results from our first simulation study indicate that the model selection methods SLOBE, mBIC2 and adaptive LASSO have much better predictive properties than the methods based on single marker tests and the least-squares approach based on all available genetic data. Single marker tests are very inefficient when the number of causal variants, $k$, is moderate or large, while the least squares approach works badly when $k$ is small. We could observe that mBIC2 and SLOBE have predictive properties similar to the ones of adaptive LASSO, with mBIC2 performing the best (having the largest precision in estimating the predictive index and identifying responsive patients) when $p$ is larger than $n$. It is important to note that SLOBE and mBIC2 achieve these good predictive properties using much less biomarkers than adaptive LASSO, which selects many uninformative SNPs.

Our second simulation study based on SNPs having a block correlation structure as one would find in GWAS yielded quite similar results. Single marker tests were again not competitive to detect prognostic and predictive biomarkers in this setting, where as previously marginal tests with Bonferroni correction resulted in a severe loss of power to detect markers, whereas the Benjamini Hochberg procedure now suffered from a hugely inflated type I error rate. The adaptive LASSO again had a much larger false discovery rate than SLOBE and mBIC2, but it remained competitive in terms of prediction and even performed best for small effect size.

The comparison between SLOBE and mBIC2 depends on the ratio between $p$ and $n$ and the number of true predictors. When $p \gg n$, the regressors are roughly independent and the number of true predictors is small, then the least squares estimators of regression coefficients in the relatively small models searched by *bigstep* have a small variance and the *bigstep* search strategy performs very well. In this case mBIC2 performs better than SLOBE, whose estimates for all $p$ regression coefficients have a relatively larger variance. mBIC2 continues to work very well to detect sparse models when correlations between regressors have a block structure like SNPs in GWAS due to linkage disequilibrium. Thus mBIC2 seems to be a better (maybe optimal) tool to identify important SNP biomarkers from GWAS. However, the relative comparison between mBIC2 and SLOBE might look different when the ratio of $p/n$ is smaller or when the

regressors are substantially correlated. In these situations SLOBE might provide a more stable FDR control, larger power and enhanced prediction properties as compared to mBIC2 [27]. Thus, SLOBE might be a better tool to identify biomarkers based on highly correlated gene expression or proteomic profiles, which remains to be tested in future studies.

Another interesting issue worth of exploring is the design of the sub-population efficacy test procedures. In our simulations we used half of the patients to select biomarkers and construct a model to identify responders and another half to test for the treatment efficacy. It seems however that the optimal sizes of these two samples should depend on the ratio between the number $p$ of covariates which need to be searched through to build the optimal model and the sample size $n$. When $p \gg n$ then constructing a good model becomes much more challenging than the testing procedure used in the second step. Therefore, it seems plausible to assign more patients to the first group. The issue of the optimal sample and alpha splitting is an interesting topic for further research.

Method 4 allows for testing the treatment effect in both the full study population and in a biomarker defined subset. By using a simple Bonferroni split the type I error rate is controlled. Such a strategy could be easily embedded in an adaptive design [49] where potential candidates for predictive biomarkers are available at the start of the trial, but there is still uncertainty on the model to determine a patient's predictive index. The sample splitting used in our simulation study corresponds to a clinical trial with an interim analysis halfway through. Then the model for a patient's predictive index can be built at an interim analysis. When testing whether there is a treatment effect in the full population all patient data can be used, whereas for the hypothesis in the biomarker targeted subgroup only data collected after the interim analysis which have not been used for developing the biomarker model. For clinical trials with adaptive interim analysis, several designs have been proposed which allow for flexible strategies to spend the $\alpha$ levels [14, 50]. However, usually these type of designs have a pre-defined subgroup, whereas here the subgroup depends on the first stage data. Furthermore the statistics to test overall efficacy in the full data set and efficacy in the subgroup of responders are positively correlated. Therefore the Bonferroni correction used in our Method 4 will be conservative and there exist techniques to improve upon Method 4 [51, 52]. The main idea would be to adjust the significance level by considering the correlation structure induced by the overlapping data points. However, it has to be noted that commonly these procedures assume known error variance, which is not the case in our situation. So further research will be necessary to adapt these different methods to improve on Method 4.

Finally we want to mention that both mBIC2 and SLOBE are approximations to a fully Bayesian procedures. Specifically, they can be easily modified to include prior knowledge on potentially important biomarkers. Such prior knowledge would summarize the results of previous experiments, it would effectively increase the sample size and thus substantially reduce the problems related to large $p$ and small $n$ issues. Further extension of our methodology in this direction could be another interesting topic for further research.

## Supporting information

**S1 Appendix. Heritability.** Computation of heritability for simulation scenarios.
(PDF)

**S2 Appendix. Additional results.** Complete results from the three simulation studies (including additional Figures and Tables) are provided as an html file.
(HTML)

**S3 Appendix. SNP description.** Detailed description of the SNP data set in genereal and the specific SNPs which were used for the data generating model.
(HTML)

**S1 Data. An Rdata file which contains the SNP data which was used for the second simulation study and which also includes the cluster information obtained with the R package *geneSLOPE*.**
(RDATA)

**S2 Data. R code for all simulations which were performed.** The zip archive also includes R Markdown files which were used to obtain summary statistics from the simulation results and a README file which briefly indicates the organization of the simulation files.
(ZIP)

## Author Contributions

**Conceptualization:** Florian Frommlet, Malgorzata Bogdan.

**Formal analysis:** Florian Frommlet, Piotr Szulc.

**Funding acquisition:** Franz König, Malgorzata Bogdan.

**Methodology:** Florian Frommlet, Piotr Szulc, Franz König, Malgorzata Bogdan.

**Software:** Florian Frommlet, Piotr Szulc.

**Visualization:** Florian Frommlet.

**Writing – original draft:** Florian Frommlet, Malgorzata Bogdan.

**Writing – review & editing:** Florian Frommlet, Franz König, Malgorzata Bogdan.

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
