## [Decision Letter · Decision Letter 0]

12 Oct 2021

PONE-D-21-14217Selecting predictive biomarkers from genomic dataPLOS ONE

Dear Dr. Frommlet,

Thank you for submitting your manuscript to PLOS ONE. After careful consideration, we feel that it has merit but does not fully meet PLOS ONE’s publication criteria as it currently stands. Therefore, we invite you to submit a revised version of the manuscript that addresses the points raised during the review process, particularly the following issues: 1) simulation of more realistic SNPs that are correlated, 2) identification of prognostic and predictive biomarkers, 3) Bonferroni correction for correlated tests, 4) application of the proposed methods to real data, and 5) possible overlap with authors' previous publication (Frommlet and Bogdan, 2020).  Please ensure that your decision is justified on PLOS ONE’s publication criteria and not, for example, on novelty or perceived impact.

We look forward to receiving your revised manuscript.

Kind regards,

Xiaodong Cai

Academic Editor

PLOS ONE

Journal Requirements:

“PS and FK were supported by the European Union’s 7th Framework Programme for research, technological development and demonstration under Grant Agreement no 602552, co-financed by the Polish Ministry of Science and Higher Education under Grant Agreement 2932/7.PR/2013/2. MB gratefully gratefully acknowledges the support by the grant Nr 2016/23/B/ST1/00454 of the Polish National Center of Science”

“PS and FK were supported by the European Union's 7th Framework Programme for

research, technological development and demonstration under Grant

Agreement no 602552,

https://ec.europa.eu/growth/sectors/space/research/fp7_en

PS and FK were co-financed by the

Polish Ministry of Science and Higher Education under Grant Agreement 2932/7.PR/2013/2. 

https://www.gov.pl/web/science

MB gratefully gratefully acknowledges the support by the grant Nr 2016/23/B/ST1/00454 of the Polish National Center of Science.  https://ncn.gov.pl/?language=en”

Reviewers' comments:

Reviewer's Responses to Questions

**Comments to the Author**

1. Is the manuscript technically sound, and do the data support the conclusions?

Reviewer #1: Yes

Reviewer #2: Yes

Reviewer #3: Partly

2. Has the statistical analysis been performed appropriately and rigorously? 

Reviewer #1: Yes

Reviewer #2: Yes

Reviewer #3: Yes

3. Have the authors made all data underlying the findings in their manuscript fully available?

Reviewer #1: Yes

Reviewer #2: No

Reviewer #3: Yes

4. Is the manuscript presented in an intelligible fashion and written in standard English?

Reviewer #1: Yes

Reviewer #2: Yes

Reviewer #3: Yes

5. Review Comments to the Author

Reviewer #1: The paper is beyond my competence in mathematical statistics, but it seems to be of very good quality. It is well written and the methods are advance. To the best of my knowledge the manuscript is satisfactory, but a specialist in mathematical statistics should do a final assessment.

Reviewer #2: Summary:

This paper focuses on the selection of prognostic (main effect) and predictive (interaction with treatment) biomarkers from a large number of candidate SNPs. SLOBE and mBIC2 are compared with adaptive LASSO, and have good performance in the simulation in terms of biomarker selection and subgroup identification. The paper is nicely written with a wide range of simulation scenarios.

Major comments:

1. It is assumed all the SNPs are independent, while in reality the multicollinearity of neighboring SNPs makes them highly correlated. The Bonferroni correction is overly conservative in association studies in which the tests are correlated. Adaptive LASSO and SLOBE have advantages with correlated covariates. I am wondering what’s the performance of the proposed methods with correlated SNPs.

2. In the simulation, half of the causal variants were prognostic and the other half predictive. So there is no overlap in the biomarkers for main effect and interaction with dose. In reality a biomarker can be both prognostic and predictive. I am wondering if the proposed methods can identify both effect and what’s the performance.

3. For biomarker identification, Fig 1 doesn’t distinguish between prognostic and predictive biomarkers. However, predictive biomarkers play a role in the personalized treatment selection and are of interest in clinical settings. It would be good to know the selection of predictive biomarkers.

4. From part 1 results, mBIC2 has best performance under different scenarios and is adopted in part 2. Method 1, 2, 2a tested treatment effect in whole population, method 3 & 4 tested treatment effect in predicted responders R(X)>0, and method 5 & 6 tested treatment effect overall or in subgroup. They are different hypotheses and methods within each hypothesis are comparable. It is confusing to compare method 2 vs method 4 and put all methods in one figure.

5. When test treatment effect overall or in subgroup, each test is performed at α=0.025 with Bonferroni correction. Those two tests are correlated and Bonferroni correction might be too conservative. For the adaptive clinical trial as in the discussion, different α-spending functions can be used and should be discussed.

6. SLOBE and mBIC2 are compared with adaptive LASSO, it would be helpful to provide the equation of adaptive LASSO too and compare with equation (3) SLOPE.

Minor comments:

The resolution of the figures needs to be improved.

Reviewer #3: This paper needs to be rewritten as many terms are not introduced at all. For example, I am not sure what are ‘prognostic index’, ‘Treatment Efficacy’, etc. In addition, no real-world genomic data were used in this paper but only some simulated SNPs. This paper will be greatly improved if the authors can apply the machine learning models directly to human genomics data. There are many published human genomic datasets in the GEO database, the authors can take advantage of them.

The content of this paper seems to overlap with the authors’ other paper (Frommlet and Bogdan, 2020) titled ‘Identifying important predictors in large data bases - multiple testing and model selection’.

6. PLOS authors have the option to publish the peer review history of their article (what does this mean?). If published, this will include your full peer review and any attached files.

Reviewer #1: No

Reviewer #2: No

Reviewer #3: No

---

## [Author Response · Author response to Decision Letter 0]

18 Dec 2021

We have provided a detailed rebuttal letter which answers all the issues raised by the academic editor and the three reviewers. In our marked-up version of the manuscript text passages which were added or changed are written in blue colour.

---

## [Decision Letter · Decision Letter 1]

20 May 2022

Selecting predictive biomarkers from genomic data

PONE-D-21-14217R1

Dear Dr. Frommlet,

We’re pleased to inform you that your manuscript has been judged scientifically suitable for publication and will be formally accepted for publication once it meets all outstanding technical requirements.

Kind regards,

Fabio Rapallo, Ph.D.

Academic Editor

PLOS ONE

Additional Editor Comments (optional):

Please check the notation in Equation 4 (see the reviewer comment) and do a final check on the English language before sending your files to the editorial process.

Reviewers' comments:

Reviewer's Responses to Questions

**Comments to the Author**

1. If the authors have adequately addressed your comments raised in a previous round of review and you feel that this manuscript is now acceptable for publication, you may indicate that here to bypass the “Comments to the Author” section, enter your conflict of interest statement in the “Confidential to Editor” section, and submit your "Accept" recommendation.

Reviewer #2: All comments have been addressed

Reviewer #3: (No Response)

2. Is the manuscript technically sound, and do the data support the conclusions?

Reviewer #2: Yes

Reviewer #3: Yes

3. Has the statistical analysis been performed appropriately and rigorously? 

Reviewer #2: Yes

Reviewer #3: N/A

4. Have the authors made all data underlying the findings in their manuscript fully available?

Reviewer #2: Yes

Reviewer #3: Yes

5. Is the manuscript presented in an intelligible fashion and written in standard English?

Reviewer #2: Yes

Reviewer #3: No

6. Review Comments to the Author

Reviewer #2: The additional simulation of real SNP data is very informative and the definition of true positive detections in the highly correlated markers is important. The relaxed definition of TP is more consistent than strict definition with correlated SNPs. There is no much difference whether the SNP is prognostic, predictive, or both. mBIC2 has slightly better performance than SLOBE with faster computation time. Thank you for fully corresponding to my questions.

Equation 4, should the beta_j be b_j comparing to equation 3?

Reviewer #3: I would like to thank the authors for addressing my comments. Still, no real-world genomic data were used in this paper but only some simulated SNPs.

7. PLOS authors have the option to publish the peer review history of their article (what does this mean?). If published, this will include your full peer review and any attached files.

Reviewer #2: No

Reviewer #3: No

---

## [Editor Report · Acceptance letter]

8 Jun 2022

PONE-D-21-14217R1 

Selecting predictive biomarkers from genomic data 

Dear Dr. Frommlet:

I'm pleased to inform you that your manuscript has been deemed suitable for publication in PLOS ONE. Congratulations! Your manuscript is now with our production department. 

Kind regards, 

on behalf of

Dr. Fabio Rapallo 

Academic Editor

PLOS ONE